# A Novel Estrogen Receptor β Agonist Diminishes Ovarian Cancer Stem Cells via Suppressing the Epithelial-to-Mesenchymal Transition

**DOI:** 10.3390/cancers14092311

**Published:** 2022-05-06

**Authors:** Ananya Banerjee, Shurui Cai, Guozhen Xie, Na Li, Xuetao Bai, Kousalya Lavudi, Kevin Wang, Xiaoli Zhang, Junran Zhang, Srinivas Patnaik, Floor J. Backes, Chad Bennett, Qi-En Wang

**Affiliations:** 1Department of Radiation Oncology, College of Medicine, The Ohio State University, Columbus, OH 43210, USA; banerjee.159@osu.edu (A.B.); cai.553@buckeyemail.osu.edu (S.C.); li.10987@osu.edu (N.L.); bai.369@osu.edu (X.B.); kousalya.lavudi@osumc.edu (K.L.); junran.zhang@osumc.edu (J.Z.); 2Comprehensive Cancer Center, The Ohio State University, Columbus, OH 43210, USA; xie.guozhen@mayo.edu; 3Columbus Academy, Gahanna, OH 43230, USA; wangk23@columbusacademy.org; 4Department of Biomedical Informatics, College of Medicine, The Ohio State University, Columbus, OH 43210, USA; xiaoli.zhang@osumc.edu; 5School of Biotechnology, Kalinga Institute of Industrial Technology, Bhubaneswar 751024, India; srinivas.patnaik@kiitbiotech.ac.in; 6Division of Gynecologic Oncology, College of Medicine, The Ohio State University, Columbus, OH 43210, USA; floor.backes@osumc.edu; 7Drug Development Institute, Comprehensive Cancer Center, The Ohio State University, Columbus, OH 43210, USA; chad.bennett@osumc.edu

**Keywords:** Estrogen receptor, cancer stem cells, epithelial-to-mesenchymal transition, ovarian cancer

## Abstract

**Simple Summary:**

Estrogen receptors α (ERα) and β (ERβ) show distinct contributions to tumor initiation and progression. Given that ERβ mainly functions as a “tumor suppressor”, activation of ERβ using its specific agonist should be able to inhibit tumor progression. In this study, we show that a newly developed ERβ agonist, OSU-ERb-12, can not only impede ovarian cancer cell expansion and tumor growth but also reduce the cancer stem cell (CSC) population. OSU-ERb-12 could inhibit epithelial-to-mesenchymal transition (EMT) in ovarian cancer cells by increasing Snail expression, thereby, blocking EMT-mediated cancer cell dedifferentiation. We also found that OSU-ERb-12 inhibits ovarian cancer expansion in an ERα-independent manner while limiting the CSC population in an ERα-dependent manner. Taken together, our data suggest that the newly developed ERβ agonist OSU-ERb-12 can be used not only to hinder tumor growth but also has the potential to prevent tumor relapse and metastasis by depleting CSCs.

**Abstract:**

Epithelial ovarian cancer is the most lethal malignancy of the female reproductive tract. A healthy ovary expresses both Estrogen Receptor α (ERα) and β (ERβ). Given that ERα is generally considered to promote cell survival and proliferation, thereby, enhancing tumor growth, while ERβ shows a protective effect against the development and progression of tumors, the activation of ERβ by its agonists could be therapeutically beneficial for ovarian cancer. Here, we demonstrate that the activation of ERβ using a newly developed ERβ agonist, OSU-ERb-12, can impede ovarian cancer cell expansion and tumor growth in an ERα-independent manner. More interestingly, we found that OSU-ERb-12 also reduces the cancer stem cell (CSC) population in ovarian cancer by compromising non-CSC-to-CSC conversion. Mechanistically, we revealed that OSU-ERb-12 decreased the expression of Snail, a master regulator of the epithelial-to-mesenchymal transition (EMT), which is associated with de novo CSC generation. Given that ERα can mediate EMT and facilitate maintenance of the CSC subpopulation and that OSU-ERb-12 can block the transactivity of ERα, we conclude that OSU-ERb-12 reduces the CSC subpopulation by inhibiting EMT in an ERα-dependent manner. Taken together, our data indicate that the ERβ agonist OSU-ERb-12 could be used to hinder tumor progression and limit the CSC subpopulation with the potential to prevent tumor relapse and metastasis in patients with ovarian cancer.

## 1. Introduction

Epithelial ovarian cancer (EOC) is the most lethal malignancy of the female reproductive tract with a low five-year survival rate of only 28% in advanced stages [1]. Serous ovarian carcinomas comprise ~70% of all EOCs [2]. The most aggressive subtype, high-grade serous ovarian cancer (HGSOC), accounts for 90% of these serous carcinomas and the majority of ovarian cancer deaths [3]. 

Most tumors are initially responsive to conventional chemotherapy, and the patients enter into clinical remission after initial treatment. However, tumor metastasis and recurrence occur in more than 70% of EOC patients despite treatment, ultimately leading to death from EOC [4]. Therefore, identifying efficient ways to halt ovarian cancer progression is particularly important to improving progression-free survival and decreasing mortality in ovarian cancer patients.

Ovarian tissues express both Estrogen receptor α (ERα) and β (ERβ) [5]. While the expression of ERα remains fairly constant, the expression of ERβ gradually decreases as the cells undergo malignant transformation and further with the progression of cancer [6]. ERα is thought to promote the expression of genes associated with cell survival and proliferation, enhancing tumor growth and progression [7]. In contrast, the natural function of ERβ is anti-inflammatory and pro-apoptotic [8,9,10,11]. 

ERβ was also found to repress the transactivity of ERα and block its cellular proliferative action [12,13]. In addition, the ability of ERβ to reduce the development of orthotopic ovarian xenograft further supports the role of ERβ as a ‘tumor suppressor’ in EOC tumorigenesis [12]. Thus, the activation of ERβ by using its agonists should be able to inhibit ovarian cancer progression.

Although numerous chemically diverse selective ERβ agonists have been described [14], only a few eventually entered clinical trials; however, they did not show favorable outcomes [15,16,17,18]. The para-carborane BE120 was shown to be an effective 17β-estradiol mimic and exhibited high ERβ selective activity with sufficient druglike properties [19]. Based on BE120, Sedlak and his colleagues synthesized a series of para-carbonate ERβ agonists and demonstrated that OSU-ERb-12 (compound 8) has low nanomolar potency, greater selectivity for ERβ over ERα, limited off-target activity against other nuclear receptors and favorable pharmacological properties compared to other clinically developed ERβ agonists, such as ERB-041 and LY500307 [20]. However, its effect on ovarian cancer is unknown.

Tumors contain phenotypically and functionally heterogeneous cancer cells. Among them, a minority cell subpopulation, termed cancer stem cells (CSCs), which possess the capacity for self-renewal and are able to generate the heterogeneous lineage of cancer cells, has been believed to be responsible for the initiation and maintenance of tumors [21]. Apart from their self-renewal and differentiation properties shared with stem cells, CSCs are characterized by high tumorigenic potential and are also referred to as tumor-initiating cells (TICs) [22]. 

Thus, the eradication of CSCs could be an effective way to improve the prognosis of patients with cancer. The CSC population can be maintained in the tumor via a balanced conversion between non-CSCs and CSCs. The disruption of this balance can induce CSC depletion or expansion. The epithelial-to-mesenchymal transition (EMT) is a process through which epithelial cells acquire the mesenchymal gene program to facilitate their migration and invasion. The EMT program has been linked to the generation of CSCs from non-CSCs [23]. Thus, EMT inhibition has the potential to limit the CSC population in a tumor.

In the present study, we demonstrate that OSU-ERb-12 inhibits ovarian cancer cell proliferation both in vitro and in vivo and reduces the CSC subpopulation via the inhibition of EMT in ovarian cancer cells. Given that CSCs are responsible for tumor relapse, treatment with OSU-ERb-12 should have the potential to improve the prognosis of ovarian cancer patients.

## 2. Materials and Methods

### 2.1. Cell Culture and Reagents

The human ovarian cancer cell lines Kuramochi and OVCAR4 were kindly provided by Dr. Adam Karpf from the University of Nebraska. OVCAR3 cells were purchased from ATCC, PEO1 cells were kindly provided by Dr. Rugang Zhang from the Wistar Institute, and OV2008 cells were provided by Dr. Francois X. Claret from the MD Anderson Cancer Center. All cell lines were authenticated by DNA (short tandem repeat) profiling and tested for Mycoplasma contamination routinely. 

These cells were cultured in RPMI 1640 medium supplemented with 10% FBS, 100 μg/mL Streptomycin and 100 units/mL Penicillin. Primary ovarian tumor cells were isolated from an HGSOC Patient-Derived Xenograft (PDX-2414) provided by Dr. Jinsong Liu at the MD Anderson Cancer Center (Houston, TX, USA). The tumor was washed with RPMI-1640 medium and then minced and digested with Collagenase in RPMI-1640 medium at 37 °C for 2 h. The digested tissue was passed through a 70 µm cell strainer to obtain a single-cell suspension. Excess RBC was removed by Histopaque-1077 centrifugation. 

The live, nucleated cells collected were seeded in culture dishes and cultured in RPMI 1640 medium supplemented with 10% FBS, 100 μg/mL Streptomycin and 100 units/mL Penicillin. The ERβ agonist OSU-ERb-12 was produced and provided by the Drug Development Institute (DDI) at Ohio State University. The ERα agonist, PPT [1,3,5-tris (4-hydroxyphenyl)-4-propyl-1H-pyrazole] was purchased from Tocris (Bio-Techne, Minneapolis, MN, USA). Recombinant Transforming Growth Factor-β (TGFβ) was purchased from Millipore Sigma (St. Louis, MO, USA).

### 2.2. Establishment of Organoid Culture from Freshly Removed Human Ovarian Tumor

A freshly removed HGSOC tumor (OV39) was obtained from the Department of Pathology within 4 h after surgery in accordance with a protocol approved by the Ohio State University’s IRB. The tumor was washed with DMEM-F12 media supplemented with 5% HEPES buffer and 5% Penicillin (100 units/mL) and Streptomycin (100 μg/mL), minced and digested with Collagenase (2.5 mg/mL) in the same media at 37 °C for 20 min. The digested tissue was passed through a 70 µm cell strainer to obtain a single-cell suspension. Excess RBC was removed by Histopaque-1077 centrifugation. The live, nucleated cells collected were washed with medium, counted and seeded in pre-warmed 24-well plates in 75% Matrigel. 

The plates were then incubated at 37 °C for 15 min in an inverted position to prevent the droplet suspension from adhering to the bottom of the plate. We added 450 μL of Organoid specific media, containing advanced DMEM F12 supplemented with 1% Penicillin-Streptomycin, 1% Glutamax, 1% HEPES, 1X B27, 10 mM Nicotinamide, 1.25 mM N-acetyl cysteine, 50 ng/mL EGF, 500 nM A8301, 100 ng/mL Noggin, 100 ng/mL R-spondin 1, 10 ng/mL FGF2, 10 ng/mL FGF10, 1 μM Prostaglandin E2 and 10 μM SB202190, to each well, as described in the literature [24]. The media was changed every 3 days, and the growth of the organoids was monitored.

### 2.3. Cell Sensitivity Assay

Ovarian cancer cells were seeded in 96-well plates at densities of 5 × 10^2^ to 1 × 10^3^ cells/well for 24 h, and then treated with different concentrations of ERβ agonist, OSU-ERb-12, once at the beginning and further cultured for 7 days. The cells were then washed with PBS, fixed with 3.7% formaldehyde for 30 min and stained with 1.0% Methylene Blue for 60 min. The plates were dried after rinsing under running water. We added 100 μL of dissolving solution (10% acetic acid, 50% methanol in water) to each well to dissolve the cells. 

The absorbance was read at 609 nm. The vehicle-treated cells were set to 100% to calculate the relative cell viability. OV39 organoids were seeded at 10,000 cells/well in 75% Matrigel in a 96-well plate and treated with different concentrations of OSU-ERb-12 for 7 days. The Promega 3D cell titer Glo luminescent cell viability assay kit (Cat. No. G7571, Madison, WI, USA) was used to determine the relative cell viability following the manufacturer’s protocol.

### 2.4. RNA Isolation and qPCR Analysis

The total RNA was extracted using TRIzol reagent (ThermoFisher Scientific, Carlsbad, CA, USA), and reverse transcription was performed to generate cDNA using the Promega Reverse Transcription System (Promega, Cat. No: A3500) in a 20 µL reaction containing 2 µg of total RNA. A 0.5 µL aliquot of cDNA was amplified by Fast SYBR Green Master Mix (Applied Biosystems, Bedford, MA, USA. Cat. No: 43858610) in each 20 µL reaction. PCR reactions were run on the QuantStudio 3 Real-Time PCR system (Applied Biosystems). The primers used for the real-time RT-PCR are listed in Table 1.

### 2.5. Immunoblotting

Whole-cell lysates were prepared by boiling the cell pellets for 15 min in SDS lysis buffer (2% (*w*/*v*) SDS, 10% (*v*/*v*) glycerol, 62 mM Tris-HCl, pH 6.8 and a complete miniprotease inhibitor mixture (Roche Applied Sciences, Indianapolis, IN, USA)). Protein quantification was performed using a Bio-Rad DC protein assay kit. Equal amounts of protein were loaded and separated on a polyacrylamide gel. The proteins on the gel were transferred to a nitrocellulose membrane, blocked with 5% nonfat milk in TBST and incubated with appropriate antibodies (Table 2) at 4 °C overnight.

After extensive washing, the membrane was incubated with the goat anti-rabbit or goat anti-mouse antibodies conjugated with HPR (Millipore Sigma, Cat. No. 12-348, 12-349) for 1 h at room temperature. After washing, the protein bands were detected with chemiluminescence. The band intensity was further quantitated using Image J and normalized by the loading control. The relative protein amount was further calculated by comparing it to the corresponding control group.

### 2.6. Immunofluorescence

Ovarian cancer cells were treated with 5 ng/mL TGFβ and 10 μM of OSU-ERb-12 individually and in combination (OSU-ERb-12 followed by TGFβ). The cells were then fixed and permeabilized with 2% paraformaldehyde and 0.5% Triton X-100. After blocking with 20% normal goat serum, the cells were incubated with mouse anti-E-Cadherin (1:100) for 1 h at room temperature, washed with TBST four times and then incubated with goat anti-mouse IgG conjugated with FITC. Fluorescence images were obtained using the Revolve fluorescent microscope (ECHO, San Diego, CA, USA).

### 2.7. Transwell Migration and Invasion Assays

Transwell migration and invasion assays were performed using Corning Transwell inserts with or without matrigel (Corning Costar, Glendale, AZ, USA). Specifically, about 5 × 10^4^ PEO1 cells and 3 × 10^4^ OVCAR3 cells were seeded in the inserts in 500 μL of media, and the inserts were gently placed in each well of a 24-well plate containing 700 μL of media supplemented with vehicle control or 10 μM of OSU-ERb-12 for 72 h at 37 °C and 5% CO_2_. The experiment was performed in triplicates. The inserts containing the invaded and migrated cells on the underside of the filters were stained with Crystal Violet for 45 min at room temperature, rinsed with water to remove the excess stain, air dried, observed under microscope (40×) and counted.

### 2.8. ALDH Analysis

ALDH^+^ cells were analyzed and sorted with the ALDEFLUOR Kit (STEMCELL Technologies, Cambridge, MA, USA) using fluorescence-activated cell sorting (FACS) with a flow cytometer (BD Aria III). For each sample, one-half of the cells were treated with 50 mmol/L diethylaminobenzaldehyde (DEAB) to define negative gates.

### 2.9. CD44/CD117 Analysis

Anti–CD117-PE and anti–CD44-FITC antibodies (BD Pharmingen, San Diego, CA, USA) were used for FACS analyses. Briefly, the cells were incubated with antibodies on ice for 40 min in the dark. After washing with cold PBS, the cells were resuspended in 200 μL PBS and subjected to FACS analyses on a BD FACS Aria III flow cytometer (BD Biosciences, Franklin Lakes, NJ, USA).

### 2.10. Small Interference RNA (siRNA) Transfection

siGENOME human ESR1 ON-TARGET plus SMARTpool siRNA (Cat. No. L-003401-00-0005) and non-targeting control siRNA (5′-UUC UCC GAA CGU GUC ACG UdTdT-3′) were purchased from Horizon Discovery (Lafayette, CO, USA). siRNA Control, and siESR1 (100 nM) were transfected into cells using Lipofectamine 2000 (Life Technologies, Carlsbad, CA, USA).

### 2.11. Sphere Forming Assay

A total of 1000 cells were mixed with semisolid media (MethoCult H4100; STEMCELL Technologies) containing 3D Tumorosphere Medium XF medium (PromoCell, Heidelberg, Germany) and seeded in six-well Ultra-Low Attachment plates (Corning). Sphere formation was assessed after 12 days of culture. To evaluate the frequency of sphere-forming cells (SFCf), cells were plated in ultra-low attachment 96-well plates in a limiting dilution manner (1, 5, 10 and 20 cells/well) using FACS. Cells were cultured in the above-mentioned medium for 12 days. The number of wells containing spheres was counted, and the SFCf was calculated using the ELDA software (http://bioinf.wehi.edu.au/software/elda/index.html, accessed on 22 April 2021).

### 2.12. Animal Study

NSG mice (6–8 weeks, female; 20–25 g body weight) were obtained from The Jackson Laboratory (Bar Harbor, ME, USA). Animal care was in accordance with the institutional guidelines, and all studies were performed with the approval of the Institutional Animal Care and Use Committee at Ohio State University. To determine the effect of the novel ERβ agonist OSU-ERb-12 on tumor growth in vivo, small pieces of the ovarian papillary serous adenocarcinoma PDX tissue (TM00335), obtained from the Jackson Laboratory, were implanted subcutaneously into the flank of NSG mice. 

When tumors from all the animals were almost 1 cm, they were subdivided into three subgroups, one group being orally dosed with vehicle control, the second group with 10 mg/kg OSU-ERb-12 and the third group with 100 mg/kg OSU-ERb-12. The animals were dosed with the help of an oral gavage every day for 19 days, and the tumor sizes were measured using a vernier-caliper every alternate day. The bodyweight of the animals was also recorded. At the end of the experiment, the mice were euthanized, and tumor tissues were removed for the isolation of tumor cells, which were analyzed for ALDH activity using the aforementioned method.

### 2.13. Statistical Analysis

Descriptive statistics, i.e., the means ± SD, are shown in the figures. Two sample *t*-tests or ANOVA were performed for data analysis for experiments with two groups or more than two groups’ comparisons respectively for independent data. The linear mixed-effects model was used to compare the tumor growth rates to take account of the correlations among observations from the same animal. Holm’s procedure was used to adjust for multiple comparisons when needed. For all statistical testing, *p* < 0.05 was considered statistically significant. All tests were two-sided.

## 3. Results

### 3.1. ERβ Agonist OSU-ERb-12 Is Able to Suppress Ovarian Cancer Cell Proliferation

OSU-ERb-12 is a newly developed ERβ agonist that demonstrates highly selective ERβ activation [20]. To test the cytotoxic effect of OSU-ERb-12 on ovarian cancer cells, we treated a panel of ovarian cancer cell lines with OSU-ERb-12, cultured cells in the continuous presence of OSU-ERb-12 for 7 days and determined the cell viability. We found that OSU-ERb-12 could decrease the cell viability in a dose-dependent manner with IC50 ranging from 7.28 to 15.36 µM (Figure 1A).

In addition, we isolated primary tumor cells from freshly removed HGSOC tumor tissue and cultured them as organoids. We found that OSU-ERb-12 is also effective in the inhibition of organoid growth, with a similar IC50 (12.89 µM) (Figure 1A). These results indicate that OSU-ERb-12 exhibits cytotoxicity in ovarian cancer cells in vitro.

It has been shown that ERα plays a critical role in cancer cell survival and proliferation, and the tumor suppressor function of ERβ could be mediated via the inhibition of the ERα function [12]. To determine whether the effect of OSU-ERb-12 on ovarian cancer cell expansion is related to the expression level of ERα and ERβ, we evaluated the ERα and ERβ expression in these ovarian cancer cells at both the mRNA and protein levels. 

We identified a huge range of ERα expression but a relatively small range of ERβ expression in these ovarian cancer cell lines (Appendix A). Given that OSU-ERb-12 can inhibit cell expansion in all cancer cell lines tested in this study at a similar efficacy, these results indicate that the killing of ovarian cancer cells by OSU-ERb-12 is independent of ERα expression. In support of this, knockdown of ERα in PEO1 and OV2008 cells, two cell lines with high ERα expression, did not show any changes in their sensitivity to OSU-E Rb-12 (Appendix A).

To determine whether OSU-ERb-12 can inhibit tumor growth in vivo, we generated PDXs of HGSOC in NSG mice and treated tumor-bearing mice with two doses of OSU-ERb-12 for 19 days. We found that both 10 mg/kg and 100 mg/kg OSU-ERb-12 treatment significantly inhibited tumor growth (Figure 1B) without significantly affecting the body weights of mice (Figure 1C). Taken together, these results indicate that OSU-ERb-12 can suppress ovarian cancer cell proliferation both in vitro and in vivo.

### 3.2. ERβ Agonist OSU-ERb-12 Reduces the CSC Population in Ovarian Cancer Cells

The suppression of tumor growth might not eliminate the tumor due to the potential recurrence as a result of CSC enrichment. It has been reported that ERβ is upregulated in breast CSCs and plays a role in the maintenance of stemness properties in these cells [25]. Thus, we wanted to know whether the activation of ERβ by OSU-ERb-12 affects the CSC population in ovarian tumors while inhibiting tumor growth. ALDH^+^ cells are considered CSCs in many tumors, including ovarian cancer [26,27,28]. We isolated tumor cells from PDXs that were treated with OSU-ERb-12 or vehicle control as aforementioned and analyzed the percentage of ALDH^+^ cells in these PDXs using FACS to reflect changes in the abundance of CSCs. 

Surprisingly, we did not see any increase in the ALDH^+^ cell population in PDXs treated with OSU-ERb-12; instead, OSU-ERb-12 treatment significantly reduced the abundance of CSCs in these PDXs (Figure 2A,B). We further confirmed this finding in a panel of ovarian cancer cell lines and an HGSOC PDX cell line treated with OSU-ERb-12 in vitro. We found that OSU-ERb-12 treatment unanimously reduced the ALDH^+^ cell population (Figure 2C–F). In addition, OSU-ERb-12 was also able to reduce the CSC population characterized by CD44^+^CD117^+^ [29] (Appendix A). 

Furthermore, we assessed the self-renewal ability of these cells by determining the frequency of sphere-forming cells (SFCf) with a limited dilution assay. We found that OSU-ERb-12 treated OVCAR3 and OVCAR4 cells showed a significantly lower SFCf than vehicle control (Figure 2G,H). Taken together, these data indicate that the activation of ERβ by OSU-ERb-12 could reduce, rather than increase, the CSC subpopulation in ovarian cancer cells.

### 3.3. ERβ Agonist OSU-ERb-12 Suppresses the Dedifferentiation of Ovarian Cancer Cells

The abundance of the CSC subpopulation in a tumor can be regulated by CSC differentiation and non-CSC dedifferentiation [30]. To understand whether OSU-ERb-12 is able to promote CSC differentiation, we enriched CSCs from ovarian cancer cells by spheroid cultures and treated these cells with OSU-ERb-12. We did not notice a significant reduction in the self-renewal ability of these sphere cells (Appendix A), indicating that OSU-ERb-12 does not significantly affect the CSC properties. We then attempted to ascertain whether OSU-ERb-12 is able to inhibit cancer cell dedifferentiation. 

We used ALDH^+^ cells to represent CSCs and ALDH^−^ cells to represent non-CSCs and analyzed the conversion of ALDH^−^ cells to ALDH^+^ cells to represent cancer cell dedifferentiation (Figure 3A). We first isolated ALDH^−^ cells from OVCAR3 and OVCAR4 cells using FACS, cultured them in the presence or absence of OSU-ERb-12 for 7 days and measured the emergence of ALDH^+^ cells. 

As expected, ALDH^−^ cells can convert to ALDH^+^ cells during culture, demonstrating the existence of cancer cell plasticity and cancer cell dedifferentiation in these ovarian cancer cell lines. OSU-ERb-12 treatment can significantly inhibit such ALDH^−^-to-ALDH^+^ cell conversion in both cell lines (Figure 3B–E), indicating that OSU-ERb-12 reduces the abundance of the CSC subpopulation by compromising cancer cell dedifferentiation.

### 3.4. ERβ Agonist OSU-ERb-12 Inhibits EMT in Ovarian Cancer Cells

Cancer cells can acquire CSC properties via EMT [23,30]. Given that ERβ can serve as a keeper of epithelial phenotype and a repressor of metastasis [31], we reasoned that OSU-ERb-12 could inhibit EMT by augmenting the function of ERβ. To test this hypothesis, we treated the EOC cell lines PEO1, OVCAR3 and OVCAR4 with 10 µM of OSU-ERb-12 for different periods and determined the expression of the epithelial marker E-Cadherin (E–Cad) and an EMT transcription factor Snail. As expected, OSU-ERb-12 caused an increase in E-Cadherin and a decrease in Snail expression in all three cell lines (Figure 4A–C). These data indicate that the ERβ agonist OSU-ERb-12 is able to suppress spontaneous EMT in ovarian cancer cells.

It is well known that EMT in cancer cells residing in the tumor microenvironment can be triggered by TGFβ. To investigate whether OSU-ERb-12 can also inhibit TGFβ-induced EMT, we treated OVCAR3 and PEO1 cells with 5 ng/mL TGFβ for 6 h, followed by treatment with OSU-ERb-12 for 24 h. The epithelial marker E-Cad in these cells was visualized using Immunofluorescence. As expected, TGFβ treatment can efficiently induce EMT, reflected by reduced E-Cad signal in treated cells. However, treatment with OSU-ERb-12 can block TGFβ-induced alteration of E-Cad (Figure 4D). In addition, we also demonstrated that OSU-ERb-12 could inhibit TGFβ-induced Snail expression (Figure 4E,F), suggesting that the ERβ agonist OSU-ERb-12 might block EMT by suppressing the expression of the EMT transcription factor Snail.

The EMT can enhance the migration and invasion ability of cancer cells. To further validate the effect of OSU-ERb-12 on the EMT in ovarian cancer cells, we analyzed the ability of migration and invasion of ovarian cancer cells following OSU-ERb-12 treatment. The transwell migration assay clearly showed that OSU-ERb-12 can significantly inhibit the migration ability of OVCAR3 and PEO1 cells (Figure 4G). In addition, the transwell invasion assay also showed a significant inhibition of cell invasion when these cells were treated with OSU-ERb-12 (Figure 4H). These results further demonstrated that the ERβ agonist OSU-ERb-12 can inhibit EMT functionally.

### 3.5. ERβ Agonist OSU-ERb-12 Limits the CSC Population by Blocking the Function of ERα

ERβ can function as an antagonist to ERα to repress its transactivity [32]. Given that estrogen can enhance EMT in prostate cancer cells by activating ERα [33] and that EMT facilitates the generation of CSCs, we reasoned that ERβ agonists may limit the CSC population by inhibiting the ERα function. To test this hypothesis, we first examined whether ERα is involved in the expansion of CSCs. We treated a panel of ovarian cancer cell lines with the ERα selective agonist PPT and determined the CSC population characterized by the high ALDH activity. We found that PPT treatment can significantly increase the abundance of ALDH^+^ cells (Figure 5A,B). 

We also determined the effect of ERα activation on the self-renewal ability of ovarian cancer cells. Similarly, in addition to the changes in ALDH^+^ cells, ERα agonist PPT also significantly increased the sphere formation ability of both OVCAR3 and PEO1 cells (Figure 5C,D). Furthermore, knockdown of ERα by transfecting PEO1 and OV2008 cells with ERα siRNA inhibited the sphere formation ability of these cell lines as well (Figure 5E–H). These results indicate that ERα is critical to the maintenance of the CSC population in ovarian cancer. Next, we examined whether the ERβ agonist OSU-ERb-12 can suppress the expression and/or transactivity of ERα. 

We treated ovarian cancer cell lines with OSU-ERb-12 and determined the ERα protein level and expression of two ERα target genes, CCND1 and NRIP1 [34]. We found that OSU-ERb-12 does not reduce the expression of ERα (Appendix A). Instead, OSU-ERb-12 significantly reduces the mRNA level of ERα target genes (Figure 5I,J), indicating that OSU-ERb-12 can suppress the transactivity of ERα. These results suggest that, whereas OSU-ERb-12’s effects on cell growth inhibition is ERα independent, its effects on the CSC subpopulation appear to be linked to the ERα activity.

## 4. Discussion

ERβ has long been known as a tumor suppressor gene. Activation of ERβ by the use of selective agonists has been proven to be therapeutically beneficial for ovarian cancer [12,35]. In this study, we tested the anti-tumor function of OSU-ERb-12 in ovarian cancer and demonstrated that this compound is indeed able to suppress ovarian cancer cell expansion and inhibit tumor growth. More interestingly, our data showed that ovarian cancer cell lines with different ERα expression levels exhibit similar sensitivity to OSU-ERb-12-induced cell growth inhibition, suggesting that this tumor-suppressive function is independent of ERα.

Although the amino acid sequence of human ERα, and ERβ displays 97% sequence identity in DNA-binding domains, there is only 54% identity in their ligand-binding domain (LBD). OSU-ERb-12 is a derivative of para-carborane BE120 (compound 8) that can bind to the LBD domain of ERβ with 201-fold ERβ selectivity over ERα [20]. As a 17β-estradiol mimic, OSU-ERb-12 can specifically activate ERβ. After ligand binding, ERβ can form either ERβ/ERβ homodimers to transactivate its target genes, or ERα/ERβ heterodimers to inhibit the transactivity of ERα [36].

ERα has been associated with estrogen-dependent tumor growth and progression by promoting the expression of genes associated with cell survival and proliferation [7]. In contrast, ERβ exhibits anti-proliferative and pro-apoptotic functions [8,9,10,11]. Generally, the tumor-suppressive role of ERβ can be attributed to its suppression of the expression and transactivity of ERα, thereby, blocking the cellular proliferative action of ERα [12,13]. For example, ectopic expression of ERβ in ERα positive ovarian cancer cell line BG-1 downregulated the ERα protein level and led to an inhibition of basal and estradiol-induced cell proliferation as well as BG-1 cell-derived tumor xenograft growth [12]. 

ERβ can also suppress the expression of cell proliferation genes that are mainly driven by ERα at the transcriptome level [13]. However, the cytotoxicity of OSU-ERb-12 is independent of the expression level of ERα, indicating that activation of ERβ with OSU-ERb-12 can specifically alter apoptosis-related genes, probably via direct regulation. This is possible because the intrinsic differences in protein structure as well as their ability to recruit different transcription coactivators determine the selective biological effects of ERα and ERβ [37]. 

Indeed, it has been shown that ERβ can affect gene expression and signaling pathways independent of ERα [13]. It is also likely that, although some cancer cells may have high ERα expression levels, they may not have increased activated ERα due to limited ER ligands, and thus the anti-apoptosis features in these cancer cells may not be attributed to ERα. Thus, our results indicate that OSU-ERb-12 can also be used to treat ERα-negative ovarian cancer, which accounts for 29–57% of ovarian cancer patients [38,39].

In addition to exhibiting cytotoxicity to ovarian cancer cells, OSU-ERb-12 was also found to reduce the CSC population in ovarian cancer cells. This effect was observed in multiple ovarian cancer cell lines, primary ovarian tumor cells isolated from patients, as well as PDXs. The abundance of CSC subpopulation in tumors is mainly regulated by CSC differentiation and non-CSC dedifferentiation. The inhibition of CSC differentiation or/and enhancement of non-CSC dedifferentiation could expand the CSC subpopulation in tumors [30]. 

Our results showed that OSU-ERb-12 reduces the CSC population mainly by blocking non-CSC dedifferentiation and has little effect on the maintenance of CSC properties. Given that CSCs can be generated by EMT-mediated non-CSC dedifferentiation [23] and that OSU-ERb-12 is able to suppress both basic and inducible EMT in ovarian cancer cells, it is likely that OSU-ERb-12 limits the CSC population in ovarian cancer mainly by inhibiting the EMT.

The EMT can be mediated via ERα in multiple cancer types, including ovarian cancer [7,31,33,40,41]. In addition, we demonstrated that activation of ERα by the ERα agonist PPT can increase the CSC subpopulation in ovarian cancer cells. Thus, the maintenance of CSC abundance in ovarian cancer can be attributed to ERα-mediated EMT, at least partially. ERβ has been shown to inhibit the EMT, likely by antagonizing the expression and transactivation of ERα. 

For example, 17β-estradiol (E2) treatment can induce EMT in ERα positive BG-1 ovarian cancer cells via ERα activation, while overexpression of ERβ or treatment with ERβ agonist DPN completely inhibited E2-induced EMT in these cells [7]. Our data also showed that OSU-ERb-12 is able to suppress EMT in ovarian cancer cells, and OSU-ERb-12 is able to inhibit the transactivity of ERα. Thus, we believe that OSU-ERb-12 limits the CSC population in ovarian cancer mainly by inhibiting the ERα-mediated EMT. 

However, we cannot exclude the possibility that ERβ agonist OSU-ERb-12 inhibits EMT independently of ERα. As reported previously, ERβ is able to inhibit the EMT in prostate cancer cells by destabilizing hypoxia-inducible factor 1α (HIFα), which is independent of ERα [42]. Therefore, OSU-ERb-12 may also inhibit EMT by directly regulating the expression of other EMT-regulating genes.

It has been reported previously that ERβ plays a critical role in the maintenance of mammary stem cells (MSCs) and breast cancer stem cells (BCSCs); stimulation of ERβ by ERβ agonist, DPN, in BCSCs increased their sphere formation ability, while the knockdown of ERβ in BCSCs reduced sphere formation [25]. Our results seem to be different from this report, as we did not see a significant impact of ERβ agonist OSU-ERb-12 on ovarian CSC properties, including the expression of stem cell specific transcription factors and sphere formation ability. 

Instead, we clearly demonstrated an inhibitory effect of ERβ agonist OSU-ERb-12 on the de novo generation of CSCs from non-CSCs. The different functions of ERβ in breast cancer and ovarian cancer regarding maintenance of the CSC population could be attributed to the differences in cancer type.

## 5. Conclusions

In this study, we not only demonstrated the effective inhibition of ovarian cancer expansion by the novel ERβ agonist OSU-ERb-12 but also showed that the activation of ERβ by OSU-ERb-12 is able to deplete CSCs in ovarian cancer. Given that ERβ expression significantly reduces with disease progression, activation of the ERβ function by use of the specific agonist, OSU-ERb-12, could help limit the CSC subpopulation, thereby, eliminating the source of tumor metastasis and recurrence.

## Figures and Tables

**Figure 1 cancers-14-02311-f001:**
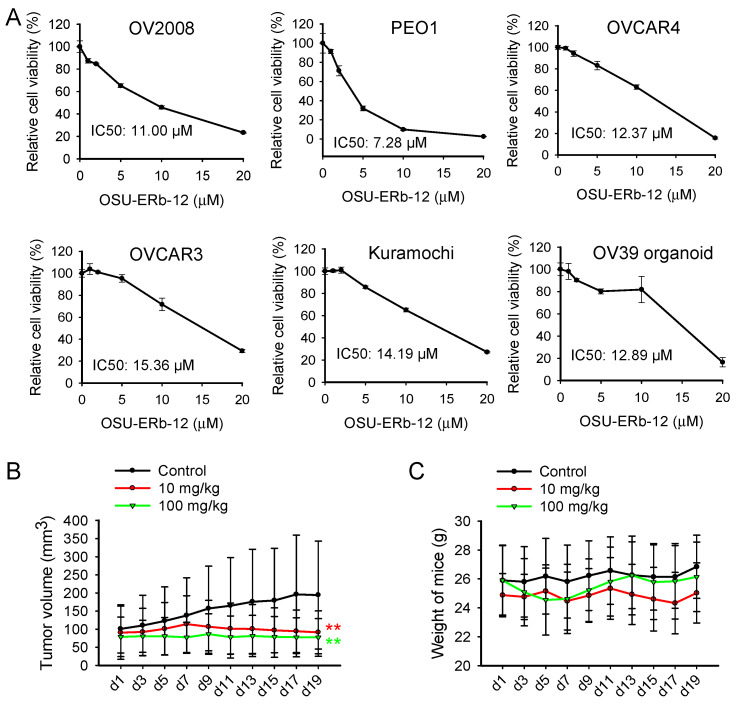
ERβ agonist OSU-ERb-12 suppresses ovarian cancer cell proliferation both in vitro and in vivo. (**A**) OSU-ERb-12 treatment reduced the cell viability of ovarian cancer cells. A panel of ovarian cancer cell lines and organoid cultured primary ovarian tumor cells were treated with OSU-ERb-12 at different doses for 7 days. The cell viability was determined using the methylene blue assay. n = 5, bar: SD. (**B**,**C**) OSU-ERb-12 suppressed ovarian tumor growth in vivo. The PDX model was generated through the engraftment of ovarian papillary serous adenocarcinoma tumor tissues (TM00335) into NSG mice. Mice were treated with vehicle control (n = 7), OSU-ERb-12 at 10 mg/kg (n = 7), or OSU-ERb-12 at 100 mg/kg (n = 5) once every day for 19 days. Tumor volumes (**B**) and mice weights (**C**) were monitored. Bar: SD. ** *p* < 0.01 compared to vehicle control.

**Figure 2 cancers-14-02311-f002:**
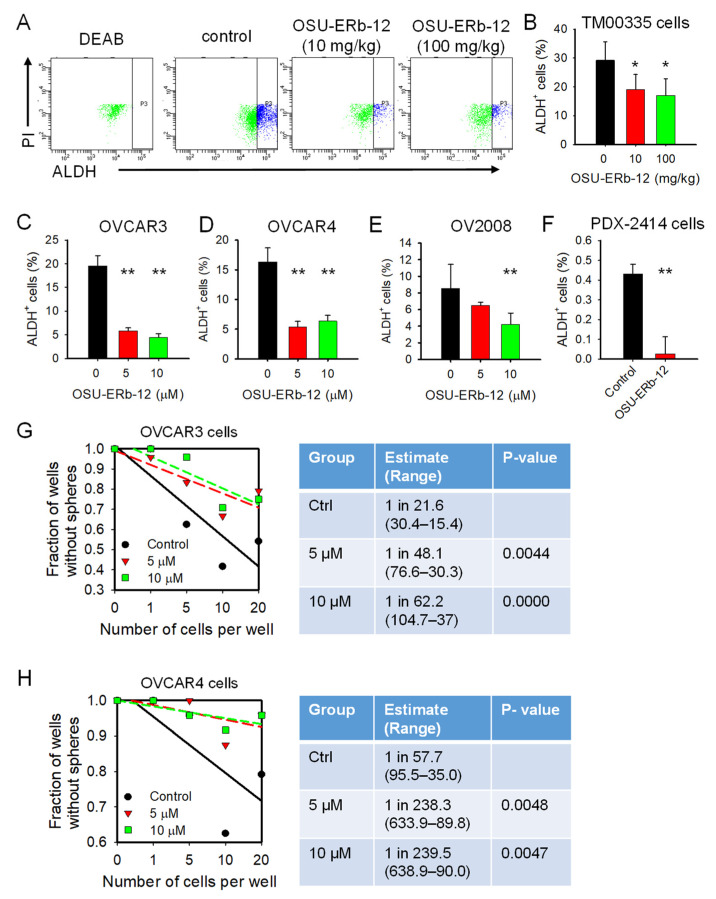
ERβ agonist OSU-ERb-12 reduces the CSC population in ovarian cancer cells. (**A**,**B**) OSU-ERb-12 treatment reduced ALDH^+^ cells in PDXs in vivo. ALDH^+^ cells in primary ovarian tumor cells isolated from PDXs described in Figure 2 were analyzed using the ALDEFLUOR assay with flow cytometry. n = 3, bar: SD. * *p* < 0.05. (**C**–**F**) OSU-ERb-12 treatment reduced ALDH^+^ cells in ovarian cancer cells in vitro. A panel of ovarian cancer cell lines and in vitro cultured primary tumor cells isolated from PDX-2414 were treated with OSU-ERb-12 for 72 h. ALDH^+^ cells were analyzed as described in (**A**). n = 3, bar: SD. ** *p*< 0.01. (**G**–**H**) OSU-ERb-12 treatment inhibited the sphere formation ability of ovarian cancer cells. OVCAR3 and OVCAR4 cells were treated with OSU-ERb-12 for 72 h; their sphere formation ability was determined using a spheroid culture with limiting dilution assay. The sphere forming cell frequency (SFCf) was calculated. n = 24.

**Figure 3 cancers-14-02311-f003:**
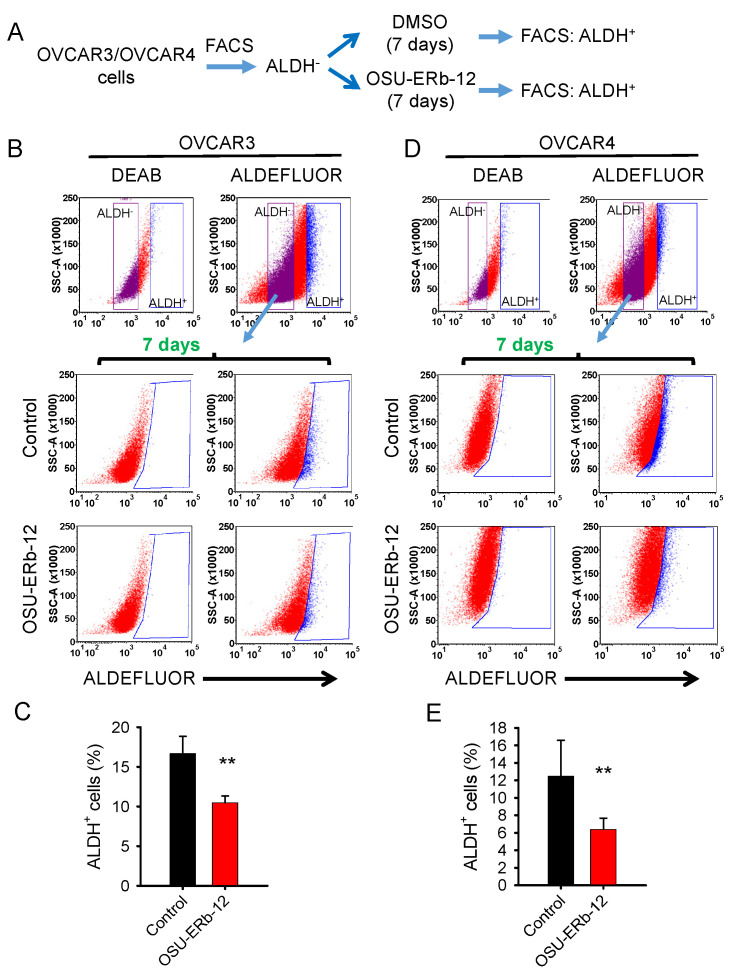
ERβ agonist OSU-ERb-12 suppresses the dedifferentiation of ovarian cancer cells. (**A**) Schematic diagram of experiments designed to analyze cancer cell dedifferentiation. (**B**–**E**) OSU-ERb-12 treatment suppressed the conversion of ALDH^−^ cells to ALDH^+^ cells. ALDH^−^ cells were sorted from OVCAR3 (**B**,**C**) and OVCAR4 (**D**,**E**) cells using the ALDEFLUOR assay and further cultured in the presence or absence of OSU-ERb-12 (10 µM) for 7 days. The percentage of ALDH^+^ cells were further analyzed using the ALDEFLUOR assay with flow cytometry. n = 3, bar: SD, ** *p* < 0.01.

**Figure 4 cancers-14-02311-f004:**
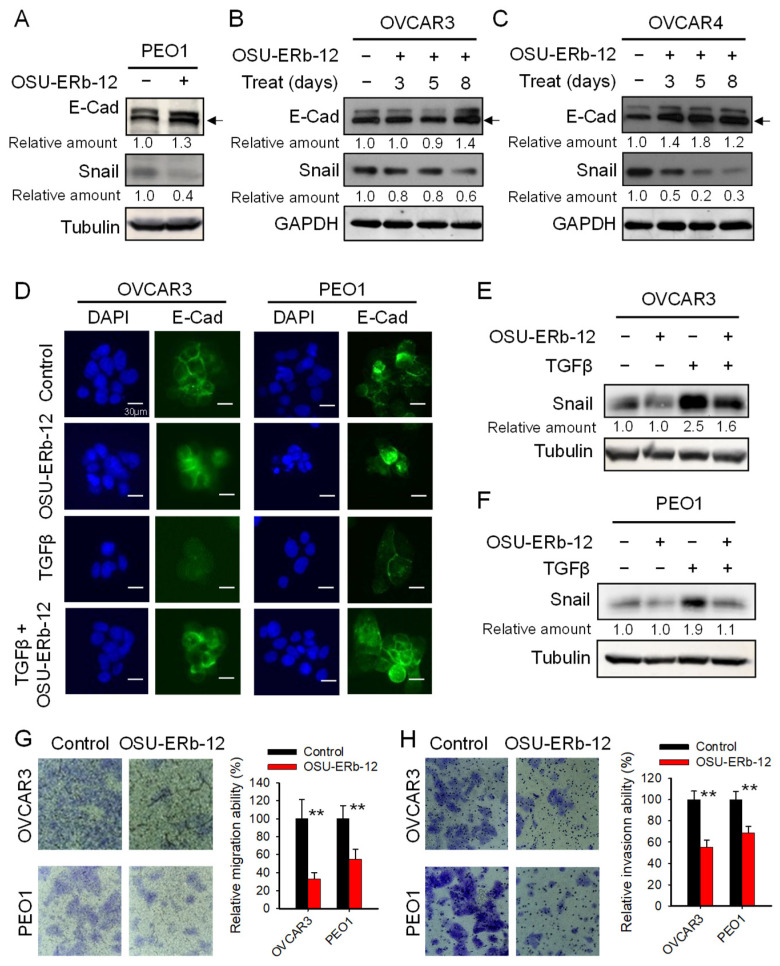
ERβ agonist OSU-ERb-12 inhibits EMT in ovarian cancer cells. (**A**–**C**) OSU-ERb-12 treatment altered expression of EMT markers. PEO1 (**A**) was treated with OSU-ERb-12 at 10 µM for 5 days, OVCAR3 (**B**) and OVCAR4 (**C**) cells were treated with OSU-ERb-12 at 10 µM for the indicated time periods, and immunoblotting was conducted to determine the expression of E-Cad and Snail. Arrow: Specific E-Cad band. (**D**–**F**) OSU-ERb-12 treatment offset TGFβ-induced EMT. OVCAR3 and PEO1 cells were treated with TGFβ (5 ng/mL) for 6 h, followed by treatment with OSU-ERb-12 (10 µM) for 24 h. Immunofluorescent analysis was conducted to determine the expression of E-Cad (**D**). immunoblotting was conducted to determine the protein expression of Snail (**E**,**F**). (**G**,**H**) OSU-ERb-12 treatment inhibited the migration and invasion ability of ovarian cancer cells. OVCAR3 and PEO1 cells were treated with 10 µM of OSU-ERb-12 for 72 h. The migration ability (**G**) and the invasion ability (**H**) of these cells were determined using the transwell migration assay and transwell invasion assay, respectively. The number of migrated or invaded cells were counted, and the relative number of migrated cells were calculated. n = 3, bar: SD, ** *p* < 0.01 compared to the control. The band intensity of the EMT proteins was quantitated and normalized by that of GAPDH or Tubulin. The relative protein amount was further calculated by comparing to the corresponding control group.

**Figure 5 cancers-14-02311-f005:**
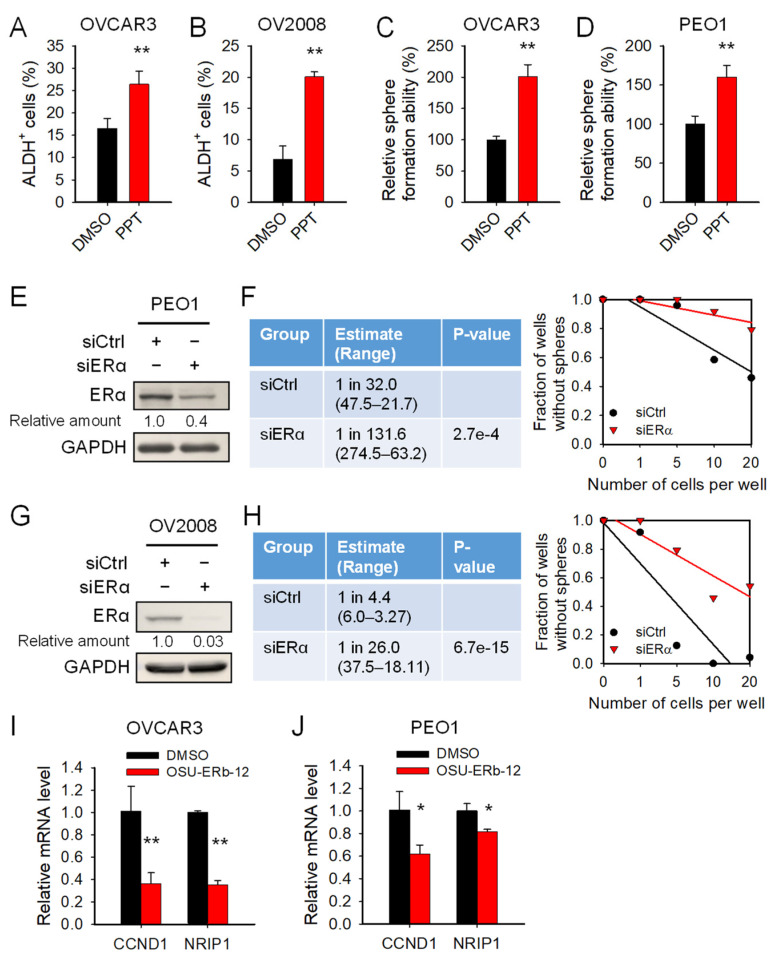
ERβ agonist OSU-ERb-12 reduces the CSC population by antagonizing the function of ERα. (**A**,**B**) ERα agonist PPT expanded ALDH^+^ cells in ovarian cancer cells. OVCAR3 (**A**) and OV2008 (**B**) cell lines were treated with ERα agonist PPT (10 µM) for 72 h. ALDH^+^ cells were determined using the ALDEFLUORE assay. (**C**,**D**) ERα agonist PPT enhanced the sphere formation ability of ovarian cancer cells. OVCAR3 (**C**) and PEO1 (**D**) cell lines were treated with PPT (10 µM) for 72 h. Their sphere formation ability was determined using the spheroid culture assay. n = 3, bar: SD. * *p* < 0.05; ** *p*< 0.01. (**E**–**H**) Downregulation of ERα reduced the sphere formation ability of ovarian cancer cells. PEO1 (**E**,**F**) and OV2008 (**G**,**H**) cells were transfected with ERα siRNA for 48 h. The expression of ERα in these cells was determined using immunoblotting (**E**,**G**). Their sphere formation ability was determined using a spheroid culture with limiting dilution assay (**F**,**H**). (**I**,**J**) OSU-ERb-12 inhibited the transactivity of ERα. OVCAR3 (**I**) and PEO1 (**J**) cells were treated with OSU-ERb-12 at 10 µM for 24 h. Quantitative RT-PCR was conducted to examine the expression level of two ERα target genes. n = 3, bar: SD. * *p* < 0.05; ** *p*< 0.01.

**Table 1 cancers-14-02311-t001:** Sequences of the primers used in this study.

Genes	Forward	Reverse
Nanog	5′-GTCCCAAAGGCAAACAACCC-3′	5′-TTGACCGGGACCTTGTCTTC-3′
Sox2	5′-TCAGGAGTTGTCAAGGCAGAG-3′	5′-GGCAGCAAACTACTTTCCCC-3′
Oct4	5′-TCGCAAGCCCTCATTTCACC-3′	5′-CGAGAAGGCGAAATCCGAAG-3′
ESR1	5′-GCTACGAAGTGGAATGATGAAAG-3′	5′-TCTGGCGCTTGTGTTTCAAC-3′
ESR2	5′-ACTTGCTGAACGCCGTGACC-3′	5′-CAGATGTTCCATGCCCTTGTT-3′
GAPDH	5′-GAAGGTGAAGGTCGGAGT-3′	5′-GAAGATGGTGATGGGATTTC-3′
CCND1	5′-GGGTTGTGCTACAGATGATAGAG-3′	5′-GAGGTGACTTCAGCCATGAATA-3′
NRIP1	5′-CTCCAAGAATGGTCTGCTAAGT-3′	5′-GGTTAAGCAAGGACCCATACA-3′

**Table 2 cancers-14-02311-t002:** Antibodies used in the study.

Antibody	Catalog Number	Company	Dilution Factor
Anti-ERα	#8644S	Cell Signaling Technology	1:500
Anti-ERß	N/A	Ohio State University	1:500
Anti-cPARP	#9546S	Cell Signaling Technology	1:1000
Anti-GAPDH	#Sc-365062	Santa Cruz Biotechnology	1:2000
Anti-Tubulin	#2144S	Cell Signaling Technology	1:2000
Anti-E-Cadherin	#610077	BD Transduction Laboratories	1:1000
Anti-Snail	#3879S	Cell Signaling Technology	1:1000

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
