# Peer review of "A Novel Estrogen Receptor β Agonist Diminishes Ovarian Cancer Stem Cells via Suppressing the Epithelial-to-Mesenchymal Transition"

_cancers, 2022, doi:10.3390/cancers14092311_

Round 1
Reviewer 1 Report
This is a nicely presented manuscript describing a novel estrogen receptor beta agonist that has the ability to reduce OVCA stem cells by targeting molecules within the EMT pathway.
However, there are a few points that can be revised to improve the paper including some typographical issues and grammatical issues in the manuscript. And also, could there be some clarification as to how the beta agonist suppresses the transactivity of ERalpha (Figure 5I), as this is not clear.
Author Response
We really appreciate reviewer’s positive comments on our study. We have carefully checked the typographical and grammatical errors and corrected them in the revised manuscript. We have also added a paragraph in Discussion to clarify the action mechanism of OSU-ERb-12 (Line 366-372).
Reviewer 2 Report
This is a interesing work that the newly developed ERβ agonist OSU-ERb-12 not only can be used to hinder tumor growth in vitro and in vivo, but also has potential to prevent tumor relapse and metastasis by depleting CSCs. ERα and ERβ show distinct contributions to tumor initiation and progression. In this manuscript both ERα and ERβ were mentioned and this is very good. But I can not get the direct connection between the activation of ERβ and the fuction of OSU-ERb-12 in ovarian cancer cells. Maybe more evidence are needed.
Author Response
We are grateful for reviewer’s positive comments on our study. The capability of OSU-ERb-12 to activate ERβ has been extensively investigated in HEK293 and U2OS cells (Sedlak D et al. J Med Chem, 2021, 64: 9330-9353). We did not further check it in ovarian cancer cells.
Reviewer 3 Report
The authors present a well conducted study. The rationale is scientifically strong and the experiments follow logical sequence. I have a few suggestions.
Major Comments:
In Fig 5, the authors use a single siRNA against ERa. A minimum of 2 siRNAs are required to rule out non-specific/off target effects. Has the effect of ERa agonist, PPT, been tested on the sphere formation? Could you please explain the rationale for this switch?
Minor Comments:
- The authors need to explain their animal model (Fig 1; TM00335). Why was this particular model used
- There is a typographical error in line 270.
- For fig2, it would be helpful to move the CD44+/CD117 expression data to the main figures. Ideally, it is best to have multiple markers to confirm the findings. It would be nicer to see this data along with ALDH data.
Best wishes.
Author Response
Major Comments:
In Fig 5, the authors use a single siRNA against ERa. A minimum of 2 siRNAs are required to rule out non-specific/off target effects. Has the effect of ERa agonist, PPT, been tested on the sphere formation? Could you please explain the rationale for this switch?
Response: We did not use “a single siRNA against ERα”. We used pooled siRNA containing 4 siRNAs targeting different sequences in the same gene. This pooled siRNA can reduce off-target silencing because the potential off-target effects of each siRNA can be diluted via competition among the siRNAs in the pool (Jackson AL, and Linsley PS. Nat Rev Drug Discov, 2010, 9: 57-67).
The effect of ERα agonist, PPT, on sphere formation has been tested. These data were presented in Figure 5C and 5D.
Regarding the rationale behind the study of the function of ERα in the maintenance of the CSC population: ERβ can function as an antagonist to ERα to repress its transactivity (Nilsson S et al. Physiol Rev, 2001, 81: 1535-1565). Given that estrogen can enhance EMT in prostate cancer cells by activating ERα (Shen Y et al. Cell Commun Signal, 2019, 17: 50), and EMT facilitates the generation of CSCs, we reasoned that ERβ agonist may limit the CSC population by inhibiting the ERα function. This rationale was included in the Result section 3.5 (Line 335-338).
Minor Comments:
- The authors need to explain their animal model (Fig 1; TM00335). Why was this particular model used?
Response: We used an ovarian tumor PDX model to investigate the efficacy of OSU-ERb-12 in inhibiting tumor growth and reducing the CSC population. This PDX was generated from an ovarian papillary serous adenocarcinoma, and obtained from the Jackson Lab, this information has been included in the manuscript (Line 217-218).
- There is a typographical error in line 270.
Response: Unfortunately, we did not find this typographical error in line 270.
- For fig2, it would be helpful to move the CD44+/CD117 expression data to the main figures. Ideally, it is best to have multiple markers to confirm the findings. It would be nicer to see this data along with ALDH data.
Response: Because not all ovarian cancer cell lines express CD44 and CD117 simultaneously, we only determined the effect of OSU-ERb-12 on the CD44+CD117+ cell population in OVCAR3 cells and used this result to support our main findings. We have already used ALDH+ as the phenotype marker and the sphere formation ability as a functional marker to characterize CSCs in the main figure. Thus, we prefer to put the CD44+CD117+ expression result in Supplementary data as supporting information.
Reviewer 4 Report
The authors have used a novel estrogen receptor beta agonist to suppress ovarian cancer by suppressing OVCAR proliferation and reducing the CSC population by preventing dedifferentiation of OVCAR ALDH- cells. Residual CSCs after the treatment is a crucial problem in cancer treatment. The authors have addressed that well in their study. However, a few concerns in the study need more explanation.
- The authors have mentioned that OSU-ERb-12 (OE12) suppressed proliferation and cell survival. Cell survival results were presented well. The authors should consider analyzing the Ki-67 expression levels (via staining or PCR) before and after OE12 treatment.
- It shows that the cell viability is reduced drastically in the presence of OE 12. Does this also infect the healthy tissue around the tumor area? Please discuss
- Both 10 and 100 mg/kg showed a reduction in the tumor volume. I am interested to know is there any saturation effect after a certain concentration?
- Following the previous comment – after removing the OE12 – do the cells regain their tumor volume? Were the mice tested for cancer recurrence and CSC presence? In other words, I’m curious to know if the OE12 exerts a permanent effect or a temporary one? Please explain
- Authors should explain the mechanism of OE12’s suppression mechanism in the discussion. It would be helpful for the readers
- EMT is more active in a confined region. Hence, please explain in which environment the EMT transition was tested.
- Please mention the sample size for each figure in the figure description
Author Response
1. The authors have mentioned that OSU-ERb-12 (OE12) suppressed proliferation and cell survival. Cell survival results were presented well. The authors should consider analyzing the Ki-67 expression levels (via staining or PCR) before and after OE12 treatment.
Response: Our cell viability assay shows a mixed outcome of both cell proliferation and cell death. Because both suppressed proliferation and increased cell death can result in reduced cell viability, we did not specifically indicate whether reduced cell viability is due to suppressed proliferation or increased cell death in this study.
2. It shows that the cell viability is reduced drastically in the presence of OE 12. Does this also infect the healthy tissue around the tumor area? Please discuss.
Response: Given that the expression of ERα remains constant, while the expression of ERβ gradually decreases as the cells undergo malignant transformation and furthers with the progression of ovarian cancer, activation of ERβ using OSU-ERb-12 should have limited effect on the surrounding normal ovary tissues because these tissues already have relatively higher ERβ activity compared to tumor tissues. Our animal study also shows that 19 days-treatment with both 10 mg/Kg and 100 mg/Kg OSU-ERb-12 did not significantly affect the health condition and body weight of mice. However, we cannot exclude the possibility that high dose of OSU-ERb-12 would damage the health tissues.
3. Both 10 and 100 mg/kg showed a reduction in the tumor volume. I am interested to know is there any saturation effect after a certain concentration?
Response: Yes, it is very likely that there is a saturation effect after a certain concentration, because the function of OSU-ERb-12 needs the expression of ERβ, and tumor cells have limited level of ERβ.
4. Following the previous comment – after removing the OE12 – do the cells regain their tumor volume? Were the mice tested for cancer recurrence and CSC presence? In other words, I’m curious to know if the OE12 exerts a permanent effect or a temporary one? Please explain
Response: We concluded our animal study at the end of the 19-day treatment period. We have not tested for tumor recurrence after drug withdrawal. As OSU-ERb-12 is able to limit the cancer stem cell population, we expect to see a delay of tumor recurrence after treatment with OSU-ERb-12.
5. Authors should explain the mechanism of OE12’s suppression mechanism in the discussion. It would be helpful for the readers.
Response: We included the putative mechanism of OSU-ERb-12’s suppression mechanism in the revised manuscript (line 366-372) as follows: “Although the amino acid sequence of human ERα and ERβ displays 97% sequence identity in DNA-binding domains, there is only 54% identify in their ligand-binding domain. OSU-ERb-12 is a derivative of para-carborane BE120 (compound 8) with 201-fold ERβ selectivity over ERα. As a 17β-estradiol mimic, OSU-ERb-12 is able to specifically activate ERβ. After ligand binding, ERβ can form either ERβ/ERβ homodimers to transactivate its target genes, or ERα/ERβ heterodimers to inhibit the transactivity of ERα.”
6. EMT is more active in a confined region. Hence, please explain in which environment the EMT transition was tested.
Response: We seeded cells at different initial numbers in a 60-mm dish, treated with OSU-ERb-12 or vehicle for 5 days (for Western blot) or 3 days (for migration and invasion). Cells were about 80-90% confluence at that time. We did not let cells grow in a confined region.
7. Please mention the sample size for each figure in the figure description.
Response: As suggested, sample size for each figure is added in the figure description.
Round 2
Reviewer 2 Report
Accept in present form.
Reviewer 3 Report
The revised manuscript is good. My concerns have been addressed and I recommend acceptance.